# Plant Species’ Capacity for Range Shifts at the Habitat and Geographic Scales: A Trade-Off-Based Framework

**DOI:** 10.3390/plants12061248

**Published:** 2023-03-09

**Authors:** Bailey H. McNichol, Sabrina E. Russo

**Affiliations:** 1School of Biological Sciences, University of Nebraska–Lincoln, 1101 T Street, 402 Manter Hall, Lincoln, NE 68588-0118, USA; bmcnichol2@huskers.unl.edu; 2Center for Plant Science Innovation, University of Nebraska–Lincoln, 1901 Vine Street, N300 Beadle Center, Lincoln, NE 68588-0118, USA

**Keywords:** climate change, demographic trade-offs, ecological strategy, geographic range, habitat filtering, habitat range, phenotypic-environment matching, phenotypic plasticity, range shift

## Abstract

Climate change is causing rapid shifts in the abiotic and biotic environmental conditions experienced by plant populations, but we lack generalizable frameworks for predicting the consequences for species. These changes may cause individuals to become poorly matched to their environments, potentially inducing shifts in the distributions of populations and altering species’ habitat and geographic ranges. We present a trade-off-based framework for understanding and predicting whether plant species may undergo range shifts, based on ecological strategies defined by functional trait variation. We define a species’ capacity for undergoing range shifts as the product of its colonization ability and the ability to express a phenotype well-suited to the environment across life stages (phenotype–environment matching), which are both strongly influenced by a species’ ecological strategy and unavoidable trade-offs in function. While numerous strategies may be successful in an environment, severe phenotype–environment mismatches result in habitat filtering: propagules reach a site but cannot establish there. Operating within individuals and populations, these processes will affect species’ habitat ranges at small scales, and aggregated across populations, will determine whether species track climatic changes and undergo geographic range shifts. This trade-off-based framework can provide a conceptual basis for species distribution models that are generalizable across plant species, aiding in the prediction of shifts in plant species’ ranges in response to climate change.

## 1. Introduction

Climate change is influencing the abiotic and biotic environmental conditions experienced by plants, with implications for species distributions [1,2,3,4]. Not only is contemporary climate change occurring more rapidly than in previous geologic time periods [5], there is also a high likelihood of the emergence of future climate regimes with no modern analog, as well as the loss of existing climate regimes [6,7,8]. These changes may cause some individuals and populations to become poorly matched to their local environments, and aggregated across all the populations of a species, this may cause changes in species’ ranges, depending on the extent of seed dispersal. However, migration rates exceeding the seed dispersal capacity of many plant species may be necessary for populations to track the environmental conditions to which they are presently adapted [9,10,11]. **Range shifts**, which we define as movement of a species’ range into new areas with potentially more suitable environmental conditions (terms in boldface font are defined in the Glossary), have been observed among plant species in the palaeoecological record in response to shorter and longer-term climatic cycles [12,13,14,15]. However, recent projections suggest that the range shifts for many plant species will lag behind the present-day rate of climate change [16,17,18,19], contributing to biodiversity redistribution globally [20].

The ranges of plant species can be defined at multiple scales (Figure 1) [21], but fundamentally are a function of the occurrence of individuals across a landscape, which is usually highly heterogeneous, because it depends on how environmental conditions are spatially and temporally distributed with respect to the ecological niche, as well as **colonization ability** (the ability to colonize sites with propagules). The **ecological niche** is the complete set of abiotic (e.g., soil, climate, topography) and biotic (e.g., mutualists and antagonists) environmental conditions allowing a plant species or population to maintain positive population growth [22,23]. How ecological niche conditions are manifested in space and time provides the template defining the habitat and geographic ranges of a species. The processes controlling fitness operate at the levels of individuals and populations responding to local environmental conditions, which we define as the habitat scale. In aggregate, the ecological and evolutionary processes operating at the habitat scale across all the populations of a species govern the distribution of a species at the largest scale, which we define as the geographic scale. Hence, the **habitat range** is the local (i.e., small spatial scale) spatial distribution of individuals of a species’ populations [24,25,26], whereas the **geographic range** is the boundary defining the geographic area encompassing the complete set of locations where those populations occur [27,28].

The capacity for shifts in habitat and geographic ranges depends on colonization ability and **phenotype-environment matching** [29,30], that is, expressing the optimal phenotype for the local environment across life stages. Phenotype-environment (PE) matching is mediated by how well the local environment matches the ecological niche [31,32,33,34], ultimately determining individual **vital rates**, and in aggregate, the population fitness in existing and novel environments [35,36,37,38,39]. Individuals that colonize a site, but have phenotypes that are mismatched with the environment, will not survive long enough to reproduce. If this consistently occurs at the population scale, then these species will experience **habitat filtering**, whereas species with phenotypes matched to the environment will establish populations. In the longer term, **adaptive evolution** allows plant populations to adjust to climate change across generations through PE matching [6,40], but rates of evolution may not keep pace with rapid climate change, particularly in long-lived plants with multi-decade maturation ages [12,41,42]. However, plants have enormous capacity for **phenotypic plasticity** [43,44,45], which, when adaptive, can enable shorter term PE matching, particularly in response to fluctuating environmental conditions, which have become a hallmark of climate change [46,47]. Phenotypic plasticity can thus allow plants to persist in their contemporary ranges, even though the climate has changed, as well as in new areas with novel environmental conditions.

Most plant species’ geographic ranges consist of a number of differently sized, spatially structured populations connected by varying rates of dispersal (**metapopulation**) [48,49,50] in areas with a range of environmental suitability (Figure 1), with the potential for **source–sink population dynamics** [51,52]. For example, many glacial remnant populations of tree species remaining in microclimate refugia may be becoming PE-mismatched sink populations with low fecundity and survival, which persist largely because of seed subsidies from source populations and the storage effect of long lifespans and vegetative propagation [53,54,55,56]. On the other hand, propagules may disperse from the range center, seeding populations at advancing range edges that, if they establish, would cause range shift or expansion.

Together, these processes lead to four broad outcomes for species’ habitat and geographic ranges under climate change. First, sufficient seed dispersal and PE matching may cause successful establishment in previously uncolonized areas, which allows plant populations to track favorable environmental conditions. Second, phenotypic adjustments, in the form of phenotypic plasticity at the individual level or adaptive evolution at the population level, may facilitate PE matching and allow populations to be maintained in contemporary ranges despite climate change. Third, sink populations caused by PE mismatch in contemporary ranges may be subsidized by seed dispersal from source populations with more favorable PE matching. Fourth, if neither colonization nor phenotypic adjustments nor source–sink dynamics occur, local population extirpation or species extinction may result. The spatial and population-scale magnitudes of these outcomes will determine whether and how species’ habitat and geographic ranges change, and specifically whether range shifts enable population persistence as the climate changes.

The goal of this review is to develop an ecological conceptual framework for understanding the probability of habitat and geographic range shifts of plant species during climate change. We focus on the first two of the above four outcomes and define a spectrum of high to low capacity for habitat and geographic range shifts by plant species. Although our focus is on changes in ranges mediated by natural processes, the expansion of non-native species in introduced ranges is governed by similar processes, and so we draw examples from plant invasion ecology. We use a trade-off-based approach to position species along this spectrum, based on ecological strategies and phenotypic trait variation. Since source–sink dynamics are likely to occur, but also likely to create complex population and range dynamics, and have been reviewed elsewhere [48,51], they are not the focus of this review. We first define habitat and geographic ranges and their scale of variation. Second, we define plant **ecological strategies**, in terms of how unavoidable trade-offs in function produce demographic trade-offs. Third, we lay out two distinct stages of processes determining the capacity for range shifts, as “arrival and survival” [57]. We translate these into colonization ability [58,59] and habitat filtering resulting from PE matching [60,61] and explain how their product determines the capacity for plant range shifts. We integrate these concepts into a trait-based conceptual model that describes combinations of traits causing variation in colonization ability and in PE matching, which together determine the capacity for range shifts in an environment. We close by identifying future research directions that will help us to better understand how plant species’ ranges will be affected by climate change at the habitat and geographic scales.

## 2. What’s in a Range? Defining Geographic and Habitat Ranges

Across spatial scales, plants can only arrive, establish, and persist in locations in which their **ecological niche** requirements are met [22,23,62,63]. Assuming no **dispersal limitation**, the mapping of this niche space onto the landscape determines the potential physical space in which a species can occur [8,64,65]. A species’ geographic range is the geographic area that encompasses the complete set of locations where a species occurs [27,28]. Geographic ranges often take the form of a system of spatially structured populations or metapopulations [48,49], because environmental heterogeneity means that not all locations within a species’ geographic range fall within its ecological niche, nor are all locations that do fall within the niche equally favorable (Figure 1). It is therefore relevant to define the habitat range [66,67] as the small-scale spatial distribution of individuals and populations, determined by where abiotic and biotic conditions are suitable [24,25,26]. As a result, the distribution, density, and demography of populations of a species are not uniform. Habitat range limits arise, where births and immigration no longer exceed deaths and emigration, and aggregated across all populations of a species, these processes define the geographic range [25,68,69,70,71] (Figure 1).

Ecological niche breadth can affect both habitat and geographic ranges. Some species have very narrow ecological niches (specialists), whereas others are found across a wide range of abiotic and biotic environmental conditions (generalists) [72,73,74]. The **fundamental niche** [22] refers specifically to the abiotic requirements of species, but as it is impossible to eliminate the effect of biotic interactions [75], the fundamental niche is a largely theoretical construct. It is therefore more useful to consider a species’ **realized niche** [22,51], which accounts for both the abiotic and biotic environment, in determining the capacity for habitat and geographic range shifts. Metrics of realized niche breadth, including stress tolerance, the availability of suitable environmental conditions, and the stability of climatic conditions through time [4,76,77,78], have been demonstrated to be associated with plant species’ geographic range sizes [4,78,79,80]. However, ecological niches may not always correspond to geographic ranges because seed dispersal can subsidize populations in sites with environmental conditions outside of the niche, and dispersal limitation can prevent populations from occupying sites with conditions inside the niche [68]. The present-day size of species’ geographic ranges may reflect the integration of these processes and therefore predict the capacity for range shifts. For example, species distribution models incorporating bioclimatic variables predicted that tree species with larger geographic ranges would have higher climatic range filling, whereas smaller-ranged species showed a stronger influence of dispersal limitation [81]. However, many “rules” of species ranges have not always been supported by empirical data [82].

The history of plant species’ evolutionary lineages and the size of the clade to which they belong also play a role in determining the species’ ranges [27,66,83,84]. However, numerous studies have assessed the influence of species’ evolutionary history on their habitat and geographic ranges [85,86,87,88,89,90], as well as the importance of **taxon cycles** in facilitating range shifts over evolutionary timescales [91,92,93]. While evolutionary and ecological dynamics have intertwined effects on range limits [71], we focus our review on the role of plant ecological strategies in shaping species’ ranges and determining species’ capacities for habitat and geographic range shifts.

## 3. Ecological Strategies and Trade-Offs

Plants exhibit a wide variety of ecological strategies, which we define as the expression of phenotypically integrated traits from molecular to whole-plant biological scales, which determine the acquisition of resources from the environment and how those resources are allocated endogenously to different functions [94,95,96,97,98]. Trade-offs in function and the need for **phenotypic integration** constrain the phenotypic variation defining these strategies to a more limited set of permissible combinations, relative to those that are possible [43,99,100]. Ultimately, trade-offs in function produce demographic trade-offs manifested at higher levels of organization (Table 1), because allocation of endogenous resources is a zero-sum game [101,102]. For example, the resource economic spectrum (Table 1) defines a productivity-durability trade-off in the construction cost of organs, such as leaves and stems. The greater investment in construction required to build physically and chemically well-defended organs is advantageous because of greater damage resistance, but this advantage comes at the cost of lower productivity and growth, due to the greater mass investment required per unit volume. This is an example of a trade-off in function that is thought to underlie a demographic trade-off of higher plant survival at the cost of reduced growth (growth-mortality trade-off; Table 1) [96,101,103,104]. Plant species falling at different points along the growth–mortality trade-off axis tend to be found in different types of environments, because more stressful environments (e.g., infertile soils) favor investment in defense and other functions promoting survival, whereas more productive environments (e.g., fertile soils) favor investment in functions supporting growth [105].

Due to numerous types of trade-offs in function (Table 1) and the need for phenotypic integration, the phenotypic traits of plant species exhibit covariation across life stages. An example comparing large and small seeded tree species is instructive. Large seeds with storage cotyledons promote survival of tree seedlings in the low-light conditions of the forest understory, and so tree species with large seeds are often tolerant of shade. Limited allocation to reproduction and seed size–number and fecundity–tolerance trade-offs (Table 1) also mean that although fewer larger seeds are produced [106,107,108,109], they tend to have a higher seedling survival. As they are heavy and not easily dispersed by wind, they also tend to be fleshy fruited and dispersed by animals [110,111,112]. Conversely, small seeds are more easily dispersed by the wind and have little room for storing resources, and so they have photosynthetic cotyledons that favor survival of seedlings in high-light environments. Small-seeded tree species thus tend to be intolerant of shade and their high fecundity facilitates colonization of rare canopy gaps where insolation is high [57], resulting in a colonization–competition trade-off (Table 1). There are certainly exceptions to these types of trait “syndromes” defining plant species ecological strategies [113,114,115], but as they are principally defined by nearly unavoidable trade-offs in function, they have implications for plant species’ capacity for shifts in habitat and geographic ranges. In the following sections, we describe how these strategies and underlying trade-offs affect processes influencing the capacity for range shifts at two plant life stages [57]: colonization ability and the likelihood of habitat filtering mediated by PE matching across life stages (Figure 2).

## 4. Colonization Ability

For shifts in plant species’ habitat and geographic ranges to occur, colonization of a new, previously unoccupied area or habitat is necessary. Following the literature on colonization–competition trade-offs (Table 1) [59,116], we define colonization as seed arrival to a site. As they are sessile, plants are notoriously dispersal limited [117,118]. Colonization limitation can prevent species from occupying suitable environments, thereby limiting the potential for populations to migrate, to follow their ecological niche during climate change (Figure 1) [9,39,65,119]. Colonization is so essential that programs in assisted migration are an attempt to overcome its limitations and facilitate plant species’ climate-driven range shifts [120,121]. Colonization also influences ecological and evolutionary dynamics within and between plant populations, and hence source–sink dynamics [51,65,122]. Aside from vicariance (i.e., fragmentation of a continuous distribution caused by dispersal barriers), rare very long-distance dispersal events [27,123,124,125,126], the spatial extent of seed rain and number of seeds dispersed far distances (the “fat tail” of the seed dispersal curve) are known to affect the rate of plant geographic range shifts and the ability to colonize new areas and habitats [14,57,127,128]. Plant traits influencing colonization ability also evolve in spatially and temporally variable environments, which can facilitate range shifts during climate change [122].

**Table 1 plants-12-01248-t001:** Summary of the key whole-plant and trait-specific trade-offs in woody species’ ecological strategies, and implications of trade-offs for the capacity for range shifts at habitat and geographic range scales.

Trade-Off	Combinations of Traits or Strategies	Effect on Habitat and Geographic Ranges	Citations
Growth–mortality trade-off (the fast-slow continuum)	Fast growth and short lifespan, or slow growth and long lifespan	Fast-growing individuals would be favored when colonization ability is more limiting and rapid climate shifts could favor a shorter lifespan (i.e., generation time)	[96,101,129,130,131,132]
Growth–defense trade-off	Greater investment in defensive compounds and/or carbohydrate storage, or in growth	Well-defended individuals with more carbohydrates (greater stress tolerance) should be less likely to be filtered out of a site than fast growers when habitat filtering is more limiting	[73,130,133,134,135]
Growth–maturation trade-off (or stature–recruitment trade-off)	Greater investment in sapling and adult stem growth and survival, or in reproduction at a younger age and smaller size (greater seedling recruitment and more years of reproduction)	Individuals with higher adult and sapling growth and survival should be less likely to be filtered out; otherwise, higher reproductive investment may be more beneficial	[129,136,137]
Seed size–seed number trade-off	Produce and disperse many small seeds or few large seeds	Smaller seed size and higher fecundity should increase colonization ability when colonization is more limiting, increasing capacity for range shift	[106,107,109,138]
Colonization–competition trade-off	Better colonizer (early arrival, fast growth, high fecundity); or better competitor (late arrival, high survival, low fecundity)	High fecundity may increase range shift capacity through increased colonization ability, but high survival may more effectively reduce probability of being filtered out	[58,116,139,140,141]
Tolerance–fecundity trade-off	Produce more, less stress tolerant seeds (better colonizer), or produce fewer, more stress-tolerant seeds (better competitor)	High fecundity may increase range shift capacity through increased colonization ability, but higher stress tolerance may reduce probability of being filtered out	[142,143,144,145]
Specialist–generalist trade-off (low or high ecological niche specialization)	Express ecological niche specialization (i.e., in abiotic conditions and relationships with mutualists and/or antagonists), or exhibit low ecological niche specialization (i.e., wider range of abiotic conditions and generalist relationships with mutualists and/or antagonists)	Higher specialization in abiotic conditions and/or with mutualists should increase the probability of being filtered out of a site; higher specialization with antagonists may decrease probability of being filtered out through reduced conspecific negative density-dependence	[85,146,147,148]
Acquisitive–conservative spectrum (leaf economics spectrum, wood economics spectrum)	Build low-cost, short-lived, thin leaves, and low-density, low- fiber wood with small carbohydrate storage; or high-cost, long-lived, thick leaves, and high-density, high-fiber wood with large carbohydrate storage	Acquisitive traits may increase an individual’s probability of being filtered out of a site when habitat filtering is more limiting, reducing capacity for range shifts	[103,104,149,150,151]
High versus low phenotypic plasticity	Express highly plastic, more costly phenotype, or less plastic, less costly phenotype	Individuals with high phenotypic plasticity should be less likely to be filtered out of a site than less plastic individuals when habitat filtering is more limiting	[29,42,43,152,153,154,155]

The spatial pattern of seed rain depends on the abundance and distribution of adults, and the shape of seed dispersal curves (probability density functions describing the probability of a seed landing a given distance from the mother tree), which is primarily determined by three axes of variation in traits related to seed dispersal: seed size, fecundity, and dispersal mode (the agents by which seeds are dispersed and their spatial movements) [111,112,117,156]. A fourth axis can be added that enables seed dispersal in time, that of seed dormancy, which has been shown to trade-off with dispersal in space [157]. These traits are frequently correlated among plant species, often referred to as seed dispersal syndromes [113,158]. Ultimately, it is the covariation in seed size, fecundity, dispersal mode, and seed dormancy defining species’ dispersal syndromes, combined with the abundance and distribution of adults, that influence colonization ability (Figure 2).

Seed size has been viewed as an organizing trait for understanding variation in colonization ability, because it is correlated with other properties of plants influencing dispersal and seedling establishment (e.g., fecundity). Across plant species globally, seed sizes vary 10^11^-fold, and among coexisting species in an environment, seed mass commonly varies over five to six orders of magnitude [159,160,161]. Although the variation between species is much more dramatic, there is also within-species variation in seed mass [162,163]. The size of a seed corresponds to maternal resource allocation per offspring, with greater mass indicating greater resource investment per seed [108,158,162].

The number of opportunities for a species’ population to disperse seeds to suitable sites is determined by the total seed production of all adults. The total seed production of a species’ population is determined by individual-level fecundity (often size-dependent), the age of reproductive maturity, and longevity [164,165]. Some species have fewer opportunities for dispersal, with a higher probability of success for each (e.g., large, high-resource investment seeds), or more opportunities, with a lower probability of success per propagule (e.g., small, low-investment seeds). This unavoidable seed size–number trade-off (Table 1) [106,107,109,138], whereby species produce few large or many small seeds, results in smaller-seeded species often being more fecund [106,107,166,167], increasing opportunities for dispersal and enhancing species’ colonization ability [168].

Beyond trade-offs with fecundity, seed size can also influence the probability of seed survival (discussed in Section 5) and secondary dispersal [108,169,170,171]. Trade-offs in the capacity for seed dispersal associated with producing smaller versus larger seeds contribute to the maintenance of seed size diversity in plant communities [144,166]. Though larger seed size can confer tolerance and higher survival when seeds are dispersed to stressful environments [142,144], there is likely to be consistent selection pressure for smaller seeds, as this directly influences population fitness [161].

Pollination limitation, in which plants produce more ovules than are ultimately fertilized, can limit seed production, reducing colonization opportunities and wasting reproductive effort [172,173]. Plant phenological shifts related to climate change (e.g., early or delayed flowering) may increase or decrease individual fecundity [174,175], suggesting strong variation in seed production could occur across a species’ geographic range, which could enhance or reduce the colonization ability of specific populations. For example, the fecundity in eastern North American forests was shown to be highest towards southern edges of species’ distributions, limiting northward spread, whereas the ample fecundity and higher recruitment in western forests is facilitating tree species’ migration to higher latitudes [19]. However, long-lived, iteroparous plant species, such as most trees, may be protected against short-term constraints on reproduction by repeated opportunities for reproduction during their lifespan [176].

Both seed size and fecundity are correlated with dispersal mode [165,177]. Smaller-seeded species can be dispersed by biotic agents with a range of body sizes (e.g., ants, vertebrates) or by abiotic agents (e.g., wind, water) [128,159,176], whereas larger-seeded species are typically dispersed by birds or mammals [112,178,179]. Directed seed dispersal, e.g., by an animal with habitat preferences, can lead to disproportionate arrival of seeds to particular locations or habitats [110,111,128], which may or may not benefit the plant [110,112]. Within growth forms, smaller-seeded species often have longer average dispersal distances than species with larger seeds [125,126,177], and may therefore have a greater colonization ability, due to their higher fecundity. However, larger-seeded species dispersed by larger-bodied animals with wide home ranges can also be carried over long distances [110,112,180], even tens of kilometers [125]. Pleistocene migration rates of large-seeded, animal-dispersed species can be comparable to those of smaller-seeded, wind-dispersed species [128], perhaps because larger seeds often survive better [108,139,178], as discussed in Section 5. Although rare, long-distance dispersal events are particularly influential in range shifts and expansions, as they enable the colonization of novel sites [126,156,181].

Beyond dispersal in space, some plant species undergo dispersal in time via seed dormancy, arising from physiological mechanisms that prevent germination when environmental conditions are unfavorable for seedling establishment [182]. Seed dormancy is often observed in particularly stressful environments (e.g., deserts), in which the favorable window is unpredictable and short [183,184], and in closed-canopy forests among light-demanding species, as the appearance of canopy gaps is rare and unpredictable [128,185,186]. Water availability, temperature, and light quality are key environmental cues affecting the breakage of seed dormancy [187,188]. Plant species with dormant seeds often have more limited spatial dispersal, representing a trade-off in dispersal in time and space [157], although this may not be true for small-seeded, light-demanding tree species. As large seeds are attractive to granivores, they often do not exhibit dormancy [157,189].

Seed dormancy allows for the establishment of a population-level seed bank at a site, enabling germination at different times within a year or in future years when conditions become suitable [182,189,190]. Given the current pace of climate change [5], the climate at a location may have changed appreciably during a period of long seed dormancy, which could be beneficial or detrimental for germination. For example, while seed dormancy release is expected to occur more quickly in response to warming temperatures, this accelerated timing may or may not correspond to periods with sufficient water availability [187], potentially negating the benefits of dormancy. The current environmental conditions of a novel site, in conjunction with other attributes of the seed, will determine whether dormancy is a strategy that enhances or limits a species’ colonization ability.

In summary, the ability of a plant population to produce ample, well-dispersed seeds is critical in determining its capacity for shifting its habitat and geographic ranges to track favorable environmental conditions. This is particularly important to consider with respect to the responses of vegetation to climate change, because the pace of migration of climate envelopes is considered to be more rapid than the migration rates of most plants, even ignoring post-dispersal survival [12,14,39]. Early-successional species with greater dispersal abilities are predicted to track climatic shifts via range shifts more readily than mid- to late-successional species, which are predicted to experience range shifts more slowly, as a consequence of reduced dispersal [191]. However, whether seeds colonize novel habitats with suitable conditions for establishment is crucial [128,168,192], as this directly affects the proportion of seeds that will survive and progress to future life stages, shaping the local population trajectory.

## 5. Phenotype–Environment Matching

After seeds successfully colonize a site, PE matching will determine whether a plant species’ population will become established in a new area (Figure 1). PE matching occurs at all life stages, including seed survival and germination, seedling survival, and maturation to a reproductive adult. Abiotic (i.e., soil, topography, climate) and biotic (i.e., mutualists and antagonists) factors and phenotypic plasticity mediate a plant’s phenotypic expression across life stages. These post-dispersal processes are critical in determining habitat and geographic range expansion and shifts. This is underscored by the fact that species that are released from dispersal limitation (e.g., species introduced by humans into a non-native range) and become invasive often have phenotypic and demographic traits that enable rapid population growth, provided the environment sufficiently matches their niche requirements, and many of these traits are also shared by native invasive species [193,194,195,196,197]. However, if a sufficiently strong PE mismatch occurs at any life stage, individuals may not survive and if this occurs at the population level, then the species will be excluded from the site via habitat filtering [198,199,200,201]. The terms “abiotic filtering”, “environmental filtering”, “habitat filtering”, and “ecological sorting” have been used in studies of community assembly and coexistence to describe the mechanisms by which plant species are excluded or “filtered out of” an environment, with nuanced differences in their definitions [105,202,203,204,205,206]. Here, we define habitat filtering, as in Maire et al. (2012) [200], to include any abiotic and/or biotic factors preventing the establishment and persistence of a particular species at a particular site. At one geographic range margin (e.g., higher latitudes in temperate regions), species’ distributions are thought to be more strongly limited by abiotic stress, whereas at the opposite range margin (e.g., lower latitudes), increasing interspecific interactions may limit distributions [27,28,207,208], but both are important in limiting geographic ranges [209]. While interactions between abiotic and biotic factors influence plant PE matching [96,209,210,211] and the potential for species’ habitat and geographic range shifts to occur, we discuss their effects separately here.

### 5.1. Abiotic Environment

A plant’s abiotic environment is principally shaped by soil properties, topography, and climate, which affect the availability of the key abiotic resources required by all plants: light, water, and nutrients [31,212,213,214]. Since plants compete for these resources and often associate with symbionts to acquire them, their availability is also moderated by biotic interactions, which will be addressed in Section 5.2. Aside from resource availability, abiotic environmental factors affect physiological and metabolic processes at all life stages and from molecular to organ-levels, such as the breaking of seed dormancy and photosynthetic carbon assimilation, as well as the expression of phenotypic plasticity, to determine plant growth, survival, and reproduction. Soil properties, topography, solar radiation, and climate differ among species and among biogeographic regions in their relative importance for defining habitat and geographic ranges [215,216,217].

Many properties of soil, including nutrient concentrations, texture, and pH, can act as local habitat filters, limiting plant distributions, often through competition for limited resources [105,199,200,218]. Soil properties can also have direct negative effects on non-adapted plant species, as in the case of serpentine soils or soils with extreme pH [219,220]. Habitat filtering mediated by soil properties and topography can limit habitat ranges. Numerous studies have found plant species to specialize on particular soil types and topographic environments [221,222,223,224], and to exhibit variation in vital rates across them [105,225]. These patterns often owe to less favorable abiotic conditions at range edges [68], limiting range shift and expansion. For example, lower soil moisture has been shown to strongly reduce germination and limit establishment towards species’ range edges [226,227].

Topography can cause heterogeneity in abiotic conditions at a range of scales [228,229], influencing the availability of belowground resources and light, as well as climate [31,230,231]. Local topographic gradients (e.g., in slope, aspect) may result in much more abrupt changes in abiotic conditions compared to broad environmental gradients (e.g., latitude) at the scale of species’ geographic ranges [25,232,233]. The rate of change and degree of variation in abiotic conditions along topographic gradients is crucial, as many plant species, particularly long-lived ones, tend to be relatively fixed in the range of climatic conditions they can tolerate [234]. As a result, microclimate refugia created by topography, rivers, and other landscape features can allow populations to persist in otherwise unfavorable regions [235]. Historically, these refugia have played important roles in defining geographic range limits, supporting remnant populations persisting outside of the core part of the range (Figure 1) [76,235,236]. Plant species with less plastic climatic niches are arguably more likely to exhibit range shifts, provided the dispersal is not limiting. For example, in warming mountainous regions across the globe, there is abundant evidence of plant populations exhibiting range shifts or contractions to higher elevations that are cooler [237,238,239]. While plant species with extensive dispersal would be expected to arrive at upslope habitats where PE matching may be better maintained [191,192], less well-dispersed species may not be able to track favorable abiotic conditions, potentially leading to local extirpation.

The rate at which plant species’ range shifts occur along elevational gradients, to track favorable thermal and moisture conditions, will likely depend on their degree of abiotic environment specialization. The degree of variation in abiotic conditions that a plant species can tolerate corresponds to a specialist–generalist trade-off, or a trade-off in low versus high ecological niche specialization (Table 1) [85,146,147], whereby species tend to have either low or high specialization to particular abiotic conditions. This ecological niche specialization also extends to plant species’ relationships with mutualist and antagonists, which we discuss in Section 5.2. The degree of specialization to abiotic conditions may directly shape species’ performance along environmental gradients. For example, a reciprocal transplant experiment evaluating seedling performance along a climatic gradient found a higher survival and growth of specialists than widespread generalist species, when planted in the specialists’ home habitat [32]. This supports the idea that widely distributed (generalist) species with large geographic or elevation ranges may have higher stress tolerance but lower competitive ability [224,240], a direct consequence of low versus high ecological niche specialization.

As average climate shifts and climatic variation increases, the proportion of surviving seedlings determines whether populations will expand, contract, or maintain their current ranges [35,74,241], which will be influenced by changes in the rate and timing of seed germination (covered in Section 4). Plant species with larger seeds generally produce more robust seedlings with higher survival (i.e., more competitive), whereas small-seeded species tend to have higher colonization abilities, reflecting a colonization–competition trade-off (Table 1) [116,168,178]. However, more severe drought events and soil warming are likely to negatively influence seedling emergence and survival [187,242], decreasing the favorability of novel sites and increasing the probability of PE mismatches. For tree seedlings, the effects of drought and climate warming on survival and growth can be buffered considerably by forest canopies reducing exposure and temperatures in the understory [243,244]. However, for species colonizing novel sites that lack overstory buffering (e.g., beyond the edge of a forest canopy or in a large canopy gap produced by a disturbance event), abiotic stress tolerance may be critical for seed survival and seedling establishment [144,145]. Species that produce large numbers of seeds tend to invest less resources per seed in conferring stress tolerance (e.g., defenses, carbohydrates, hard seed coat), whereas species that produce fewer seeds generally have more stress-tolerant seeds with higher survival, creating a tolerance–fecundity trade-off (Table 1) [142,144,145]. Thus, while having larger seeds may be favorable for species’ survival in less favorable environments, highly fecund species may be able to avoid stressful environments through increased chances for dispersal; a trade-off that can help to facilitate PE matching for species across this spectrum.

As environmental variation often increases with spatial scale, regional climatic variation shapes the pattern and limits of species’ ranges at geographic scales [80,245], whereas local microclimate variation within habitats influences the environmental conditions that individual trees experience [246]. Therefore, microclimate directly affects individual vital rates, which translates to effects on population fitness across species’ geographic ranges [25,247]. For example, some boreal tree species exhibit fitness trade-offs between growth rates and freezing tolerance in populations across their geographic ranges, limiting their ability to grow quickly and resulting in these species being outcompeted at their southern range limits by species with faster growth rates [248]. A study of 19 tree species in a North American temperate forest found differences in growth rate for populations at their range limit versus range center, with growth rates generally being lower at the colder northern limit and higher at the warmer southern limit [38]. Alternatively, species with geographic range limits imposed by climatic gradients may experience increased survival and/or reproduction at a range edge in response to increasing temperatures [41], potentially allowing for expansion of their range limits into favorable novel environments. Unavoidable resource allocation trade-offs result in species generally increasing investment in sapling and adult growth and having a later reproductive age, or prioritizing maturation, with an earlier reproductive age and more years of reproduction, representing a growth–maturation trade-off (Table 1) [141]. Indeed, plant populations at their warmer range limit have been shown to undergo compensatory shifts in their demographic rates in response to warming conditions, experiencing higher growth and lower recruitment and survival (growth–mortality trade-off; Table 1) [249]. However, tree mortality has also been shown to increase towards the warmer limits of species’ geographic ranges [250], indicating that climate warming, and potentially its interactions with other stressors (e.g., pests), are causing range contractions.

In addition to lower survival, reproductive failure may also be more likely towards habitat and geographic range limits, where conditions are often at the edge of species’ ecological niche [9,68,88,251]. Climatic conditions including rainfall, change in temperature between summers, and preceding year temperature are known to affect seed production, although interannual variation in production can be considerable in long-lived species [252,253,254]. Reproductive failure and/or reduced seed production at range edges may often be a stronger factor limiting the expansion of habitat and geographic ranges compared to the survival of offspring [26,145,255], with potentially significant consequences for edge population persistence. In summary, the effects of climate change on the suitability of abiotic environmental conditions for a species will affect PE matching across habitat and geographic ranges and in any new areas that may be colonized. However, the biotic environment can moderate the abiotic environment, and these interactions can affect PE matching.

### 5.2. Biotic Environment

Biotic interactions influence plant growth, survival, and reproduction, and the magnitude of these effects often depends on the abiotic environment [256,257,258]. Many plants rely on mutualisms for essential functions, such as seed dispersal and pollination (discussed in Section 4), as well as resource acquisition (e.g., root symbionts like nitrogen-fixing bacteria, mycorrhizae). Beneficial symbioses can cause the realized niche of plants to be larger than the fundamental niche; for example, by enabling plants to maintain populations in less fertile soils than would be possible without root symbionts [148]. Biotic interactions can also be antagonistic, as in the competitive interactions between plants and the negative effects on plants of natural enemies, such as pests, pathogens, and herbivores. Since the populations of mutualists and antagonists that plants interact with are also likely to be affected by climate change [259,260], these interactions can affect range shifts in potentially complex ways [260,261,262].

A plant species’ probability of range shifts depends on the distribution and abundance of key mutualists and antagonists and whether or not their ranges are undergoing shifts in correspondence with plant range shifts [33,148,263]. Asynchrony in biotic interactions may affect plant distributions if their partners respond to climate change in different ways, resulting in a phenological mismatch [264,265,266]. In the case of pollination, plants that rely on specialized animal pollinators, as opposed to wind pollination, generalist pollinators, or self-pollination, may be particularly vulnerable to decoupled shifts in plant and pollinator ranges [266]. The absence or lowered abundance of key specialist pollinators could lead to reduced seed set, impairing colonization and causing PE mismatch [176]. On the other hand, having less specialized interactions with mutualists is considered an important trait, allowing non-native species to become established in new areas [267]. In the case of root symbioses, decoupled range shifts could result in reduced nutrient acquisition by plants. As with pollinators, this effect would be expected to be particularly acute in the case of highly specialized symbioses, such as between pines (*Pinus* sp.) and ectomycorrhizal fungi [268]. In the case of plant antagonists, decoupled range shifts could be advantageous for plant populations, by allowing for escape from natural enemies, facilitating establishment in novel sites [34,269]. For example, the grassland species *Tragopogon dubius* experienced antagonistic effects of soil pathogens in numerous parts of its established range but avoided these effects in a newly occupied part of its range [270], suggesting this species benefitted from the decoupling of the plant and antagonists’ ranges. However, plant species with either generalist antagonists or specialist mutualists may benefit less from decoupled range shifts, as novel sites may feature other limiting natural enemies or lack key mutualists, corresponding to a specialist–generalist trade-off related to species’ ecological niche specialization (Table 1) [85,147,148].

Decoupling of the ranges of interacting plant species will also affect species’ capacity to undergo range shifts, as plants encounter differences in the magnitude of competition compared to the core parts of their ranges. Encountering better or worse competitors will affect both individual vital rates and plant species’ responses to competition [271,272]. Inferior competitors may be restricted to narrow distributions on interspersed patches of habitat due to their inability to compete for space [273], with more limited capacity for range shifts than strong competitors. Increased interspecific competition where species’ ranges overlap may impede plants from shifting their ranges to track favorable climatic conditions [273,274,275], reducing their ability to maintain PE matching.

Both mutualistic and antagonistic interactions affect the ability of plants to maintain PE matching, particularly at the seed and seedling stages. Connections of seedlings to the mycorrhizal networks of adult trees strongly increase their survival and access to soil water [276], which may facilitate better establishment of seedlings at range edges. Fungal mutualists can also allow for large-scale range expansion into otherwise unsuitable environments by reducing abiotic stressors on plants [259]. However, antagonistic interactions operating at the seed and seedling stage can halt range shifts, as seed and seedling survival are strong demographic filters [277]. Seeds and seedlings will not survive if they are vulnerable to or poorly defended against antagonists [278,279]. Seed predators can dramatically reduce the number of seeds available to establish at a site [280,281]. Indeed, most seeds are consumed by granivores [105,112,279,282], limiting potential seedling recruitment and capacity for range shifts if seed predation is high compared to seed production.

Many biotic interactions operate in density-dependent ways. For antagonistic interactions, the effects on plants can be increasingly negative as the density of plants increases, which has important consequences for population dynamics [283]. For example, optimally foraging seed predators and herbivores should spend more time in denser resource patches [284], and higher conspecific neighborhood density is often associated with lower survival and growth, mediated by intraspecific competition and natural enemies [285,286,287,288]. Soil pathogens that can strongly limit seed germination and seedling survival [289] accumulate in areas of high population density [290]. Plant apparency is greater in areas of high host plant population density, which can increase rates of herbivory [291,292]. These and other similar processes result in **conspecific negative density-dependence** (CNDD) of plant vital rates, particularly growth and survival. Seed dispersal can allow offspring to escape the CNDD operating at early life stages [128]. At species’ habitat and geographic range edges, densities of individuals at early life stages and adults are often lower than in more central parts of the range [24,219]. Colonization of areas with reduced conspecific density may alleviate CNDD and promote population growth at the range edge, facilitating range expansion [34,293]. Indeed, the enemy release hypothesis may explain the invasion success of some non-native plant species [194,293,294], although, over time, negative feedbacks can develop [295]. However, growth–defense trade-offs (Table 1) may moderate the strength of CNDD. The greater investment in physical and chemical defenses often found in slower-growing plant species may allow them to avoid the negative effects of antagonists, whereas faster-growing species may better tolerate aboveground herbivory because their organs, particularly leaves, are rapidly replaced anyway [134,296,297].

On the other hand, low population density may not always be beneficial, as it can adversely affect some interactions, particularly for mutualisms. For example, pollinators are often attracted to large floral displays, occurring in areas with a high density of flowering individuals [298,299]. **Allee effects** may occur if animal pollinators are less likely to visit more isolated individuals or low-density populations, such as at a range edge, which could limit the seed set [176,300,301,302]. Changes in the density or composition of root mutualists, such as mycorrhizal fungi, can also affect range shifts [260,303,304,305]. For example, reduced densities of mycorrhizal mutualists of native plant species have been shown to enhance the population growth of invading non-native plant species [306]. For plant species with high mutualist specialization, as in the example of pines (*Pinus* sp.) and ectomycorrhizal fungi described earlier, the density of mutualists can strongly influence plant niche breadth [268], with decreases in fungal densities limiting such species’ distributions [259,303].

Ultimately, climate change will not only affect plants via direct effects on the abiotic dimensions of the ecological niche, but it will also affect the organisms with which plants have mutualistic and antagonistic interactions, likely causing changes in their population densities and ranges. Complex decoupling of the distributions of plants from their mutualists and antagonists could inhibit or promote range shifts in ways that depend on the species’ ecological strategy and unavoidable trade-offs across life stages [263,304,305].

### 5.3. Phenotypic Plasticity

Another consequence of plants’ sessile lifestyle is phenotypic plasticity. Shifting biotic and abiotic environmental conditions can induce changes in plant phenotypic traits [307,308], which, when adaptive, can result in better PE matching [154,309]. Phenotypic plasticity may be an important property that makes non-native introduced plant species become invasive [310,311]. However, the degree to which plasticity is adaptive may depend on the time lag between the environmental change and the organism’s phenotypic shift; the longer the lag, the less adaptive the change may be, increasing the likelihood of PE mismatch [12,42]. The ability of plants to respond adaptively to climate change therefore depends on their magnitude and direction of phenotypic plasticity [312,313], within constraints imposed by phenotypic integration [97,307,308], by buffering individuals against the effects of short- and long-term shifts in environmental conditions [41,247]. This buffering may be particularly important for long-lived plants with longer generation times, by allowing individuals to plastically adjust their phenotypes in response to changing conditions [207,247]. Plasticity in phenotypic expression can also operate as habitat selection, because through their growth, plants modify their own environment [152,219], which can moderate their response to climate change throughout their lifespans [10]. Despite the potential advantages, there are still costs to expressing a flexible phenotype, representing a trade-off in high versus low phenotypic plasticity (Table 1) [42,152,154]. These costs may arise from the production of plastic phenotypes, maintaining the physiological “machinery” that allows for plastic responses, or from fitness costs if a more plastic phenotype has lower survival or performance in an environment [43,60,152,314].

Phenotypic plasticity can affect the course of evolution, and hence, the long-term responses to climate change. This can happen by either facilitating adaptive evolutionary change, by “buying time” for evolution to occur [315,316,317] or constraining it, by reducing the strength of natural selection [318,319,320]. In the plane tree (*Platanus orientalis*), populations from more favorable growing environments had greater leaf trait plasticity, which enabled beneficial adjustments during experimentally induced droughts. For individuals from populations in drier, hotter climates, however, reductions in plasticity were associated with trait values more adaptive for the prevailing, more stressful environmental conditions [321]. Whether phenotypic plasticity itself can be considered a trait is controversial, but there is evidence that, when genetic variation in plasticity is present, it can be selected for and evolve in response to environmental changes [43,313,322]. However, it is ultimately both the rate of these environmental changes and the rate and type of adjustment in plants’ expressed traits that will determine whether plasticity is adaptive and allows PE matching to be maintained and the local population to persist.

## 6. Plant Species’ Capacity for Range Shifts

The numerous functional and demographic trade-offs (Table 1) form a basis for defining a conceptual framework describing how variation in ecological strategies affects the ability of plant populations and species to undergo shifts in habitat and/or geographic ranges in response to climate change (Figure 2). We consider the likelihood of range shifts occurring to be a function of two factors, represented on the axes of Figure 2: (1) colonization ability, and (2) the probability that a given genotype will be filtered out of a particular environment due to PE mismatch (i.e., chances of habitat filtering). Variation in colonization ability and in the chances of habitat filtering both depend on components of plant ecological strategy, and patterns of qualitative variation in phenotypic traits defining this dependence are shown on each axis (traits on axes of Figure 2 are in italic face font throughout this section and apply for panels a–c). Using a conceptual state-space representation with three hypothetical plant species, the color ramp in Figure 2 shows possible ways that the capacity for range shifts depends on colonization ability and the chances of habitat filtering, with this capacity ranging from low (yellow) to high (purple). The capacity for plant range shifts is equally determined by the effects of colonization ability and the probability that a species will be filtered out of a site after colonization in Figure 2a, is more limited by a species’ probability of being filtered out in Figure 2b (i.e., less favorable environment), and is more limited by a species’ colonization ability in Figure 2c (i.e., more favorable environment). Species 1 would have a moderate *dispersal distance* and *fecundity* and small to moderate *seed size*, could represent an *abiotic condition and mutualist specialist* and *antagonist generalist* (with respect to *ecological niche specialization*), experience high pressure from conspecific negative density-dependence (*CNDD*), produce less robust seedlings (*robustness of seedling*), be a less well-defended and stress tolerant individual (*defense* and *stress tolerance*), and/or exhibit low *phenotypic plasticity*. Species 2 would have low seed *dispersal distance* and low *fecundity,* would be a large-seeded species, and could represent an *abiotic condition and mutualist generalist* and *antagonist specialist*, experience minimal effects of *CNDD*, produce more robust seedlings, be a well-defended and stress tolerant individual, and/or exhibit high *phenotypic plasticity*. Species 3 would have high colonization ability, due to having a longer *dispersal distance*, smaller *seed size*, and higher *fecundity*, and similar traits on the *y*-axis as Species 2. In this section, we describe the rationale for this trade-off-based framework.

The influence of *seed size* on a species’ capacity for range shifts will vary depending on the species’ *dispersal distance* and *fecundity*, and on whether habitat filtering, or colonization ability, is more limiting (Figure 2b, c). In general, species with increased *dispersal distances* will have a greater colonization ability, increasing their capacity for range shifts across environments (e.g., Species 3). However, species with larger *seed sizes* are often animal dispersed [111,282], and so colonization into more favorable environments depends on whether the animal dispersal agents use those environments, potentially decreasing their colonization ability (e.g., Species 2). Conversely, plant species with smaller *seed sizes* are often wind dispersed, or dispersed by generalist animals (e.g., small birds) [164,176], so colonization may be less constrained by the dispersal mode, increasing the capacity for range shifts (e.g., Species 3).

The proportion of seeds that survive and establish as seedlings will strongly affect the capacity for range shifts. Seeds and seedlings are vulnerable stages of plants and are arguably more susceptible to PE mismatch, due to several mechanisms. As a result of the colonization–competition trade-off [59,178], when habitat filtering is more limiting, having a smaller *seed size* and/or less *robust seedlings* should increase a species’ probability of being filtered out, reducing the capacity for range shift (e.g., Species 1 in Figure 2b). Conversely, when colonization ability is more limiting, having a smaller *seed size* may facilitate colonization, increasing capacity for range shift (e.g., Species 1 in Figure 2c; Table 1). Owing to the seed size–seed number trade-off [106,109], small-seeded species may not be filtered out, due to having many opportunities to colonize new sites through increased *fecundity*, increasing the species’ capacity for range shift, despite a potentially lower survivorship of individuals (e.g., Species 3 in Figure 2; Table 1). However, having high *fecundity* may be less beneficial in a less favorable environment [144,145], where habitat filtering is more limiting (e.g., Species 3 in Figure 2b), since each seed has an advantage through increased *stress tolerance* (tolerance–fecundity trade-off; Table 1), reducing the probability of being filtered out and facilitating range shift (e.g., Species 2 in Figure 2). *Stress tolerance* frequently corresponds with greater investment in *defense* against natural enemies, reducing the resources available to allocate to growth (growth-defense trade-off; Table 1). Investment in *defenses* corresponds with longer lifespan [130,323], therefore reducing the probability that a species is filtered out of a site, particularly in less favorable environments (Species 2 and 3 in Figure 2). The benefits of investment in *defenses* for species’ capacity for range shifts may be greater in less favorable environments (concave-up down in Figure 2b), because tissue replacement is more costly in poor environments [133,134].

By defining the range of possible novel sites that, once colonized, are suitable for establishment, the ecological niche affects the capacity for habitat and geographic range shifts [68]. Higher *ecological niche specialization in abiotic conditions and mutualists*, and lower *specialization in antagonists* (i.e., an *antagonist generalist*), should increase the probability that a species is excluded from a site via habitat filtering, reducing its capacity for range shifts (Figure 2). Niche specialists are expected to be more susceptible to climate change [79], due to their often narrower habitat and geographic ranges. However, if environmental changes cause suitable areas to become more available, higher *specialization in abiotic conditions and mutualists* and lower *specialization in antagonists* may less strongly influence a species’ probability of being filtered out of a site (concave-down contours, Figure 2c). A species’ degree of *antagonist specialization* should also be related to the relative importance of *CNDD*: species with highly specialized natural enemies should be able to escape their adverse effects through reduced *CNDD*, decreasing their probability of being filtered out of a site (e.g., Species 2 in Figure 2). *CNDD*, while important at all size classes, has strong negative effects on seeds and seedlings and may increase a species’ probability of being filtered out of a site [282,285,286,287], reducing the capacity for range shifts (e.g., Species 1 in Figure 2). Alternatively, if the species’ colonization ability is high, propagules dispersing to novel sites where population densities are low may escape sources of negative density dependence [34,324], increasing their capacity for range shifts (e.g., Species 3 in Figure 2). Finally, by expressing adaptive *phenotypic plasticity*, species may avoid being filtered out of a site, potentially increasing their capacity for range shifts [247,325,326] (e.g., Species 2 and 3 in Figure 2). Nonetheless, the costs associated with expressing *phenotypic plasticity* (Table 1), particularly in less favorable environments, may result in a species being filtered out of a site (concave-up contours, Figure 2b).

Ultimately, the patterns of qualitative variation in expressed phenotypic trait values that we illustrated in this conceptual model define plant species’ ecological strategies, and result in the associated, unavoidable functional and demographic trade-offs. Species’ ecological strategies and the trade-offs that characterize them shape species’ colonization abilities and their likelihood of being filtered out of a site via PE mismatches, determining their capacities for range shifts into novel environments.

## 7. Synthesis and Future Directions

Climate change may dramatically reshape the distribution of diversity on Earth, particularly in combination with other anthropogenic stressors related to human population growth, such as land-use change and landscape fragmentation and degradation [5,327,328,329]. There is a need to predict the effects of climate change on species habitat and geographic ranges, particularly for species already of conservation concern [21,330,331,332,333]. Quantitative species distribution models that predict occurrence or abundance as a function of ecological niche variables using correlative formulations, based on existing distribution information (e.g., Maxent, Bioclim) [334,335] or more mechanistic formulations (e.g., NicheMapR, Ecosystem Demography Model) [336,337,338,339], have been used to predict plant range shifts under future climate change scenarios, for both rare and common species and across plant growth forms [340,341,342,343]. Mechanistic models are viewed as more generalizable, although correlative and mechanistic models have been shown to make similar predictions [344]. These complex models take various data inputs and make many assumptions, both explicitly and implicitly, and their application is complicated [345,346]. Data integration and modeling advances have expanded the types of data that can be used in species distribution models [346], and the sources of freely available diversity data are growing (e.g., Global Biodiversity Information Facility) [347]. Still, for many plant species, particularly rare species, we presently lack the data needed to fit and validate quantitative species distribution models [8]. Moreover, the predictive accuracy of these models is often not tested or only tested in limited scenarios. Recent meta-analyses have found species distribution models in many cases to have mediocre accuracy for occurrence, and even lower accuracy for other population parameters, such as abundance [348]. Moreover, others have pointed out that many species ranges are not shifting in the expected ways [349]. These observations highlight the need for reproducibility and assessment of quantitative species distribution models and their predictions [349,350].

Modeling plant species’ distributions presents several challenges, many of which are ultimately due to their sessile nature. As described above, colonization, phenotypic plasticity, and interactions with mutualists and antagonists are fundamental to much of plant population ecology yet are still not well understood or are less predictable, making them hard to integrate into mechanistic quantitative species distribution models [102]. For example, phenotypic plasticity means that the rate of a physiological process not only changes due to the environmental conditions, but that the plant can adjust aspects of its form and function to changing conditions, in ways that allow a more adaptive response to its current environment. These types of feedbacks can be difficult to constrain in mechanistic models. Modeling at larger, geographic scales and correlative models may circumvent the need to prescribe these sorts of processes explicitly [344,351]. However, models making quantitative predictions at smaller, habitat scales would ideally represent processes affecting individuals and populations more explicitly, which may make them difficult to develop and parameterize.

The trade-off-based framework that we present here for evaluating the range shift capacity of plants clearly does not enable quantitative prediction. However, it provides a readily accessible, conceptual basis for making qualitative predictions that are predicated on generally well-supported fundamental trade-offs in function, including some that have not yet been, or only implicitly, incorporated into quantitative species distribution models. Our framework also provides a set of hypotheses about the ecological processes influencing range shifts, most of which can be tested using data that are readily available for many plant species. The untestable hypotheses point to future data collection and modeling needs, for example, on the role of mutualist specialization in limiting plant species range shifts. Our framework is likely to be particularly useful for making qualitative predictions of range shift capacity for poorly known species, such as in highly diverse tropical ecosystems, in which only basic ecological and functional information may exist, and for when there is a need to predict the range shift capacity for many plant species using a similar set of criteria. Summarizing the key trade-offs and aspects of species biology that should be considered in building a quantitative distribution model, our framework can serve as a guide for model developers.

## Figures and Tables

**Figure 1 plants-12-01248-f001:**
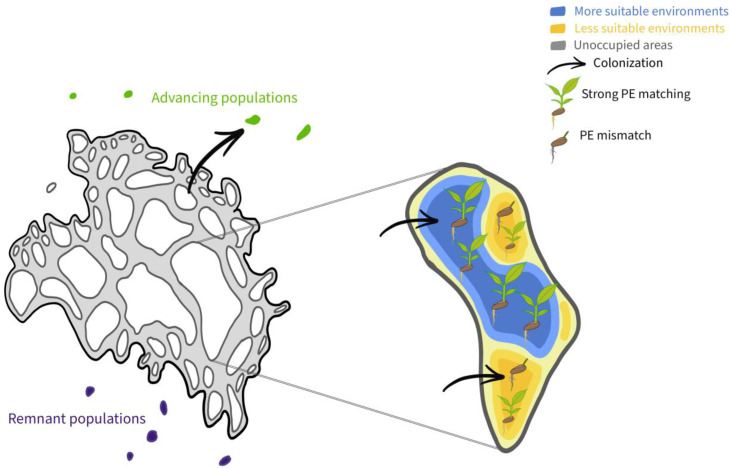
A plant species’ geographic and habitat ranges are determined by colonization and the suitability of abiotic and biotic environmental conditions with respect to the ecological niche. The left-hand portion of the figure represents the species’ geographic range as a metapopulation, in which gray shading indicates areas that are unoccupied (due to dispersal limitations or unsuitable environmental conditions), and the white polygons indicate areas that are occupied, i.e., populations of the species in suitable environments. Populations towards the geographic range edges are depicted as being smaller and populations in the range center are larger, to symbolize that geographic range edges often consist of marginal, less suitable environments, and hence smaller populations. Purple polygons depicted outside of the main geographic range represent remnant populations of the species, and green polygons represent advancing populations that have recently established in a suitable environment outside the species’ current geographic range. The right-hand portion of the figure depicts the habitat-scale distributions of one population of the species, indicating the population’s habitat range, defined by regions of more (blue) and less (yellow) suitable environments that have phenotype-environment (PE) matching or some PE mismatch (see symbols in the figure legend), thereby causing a variation in population density. Arrows on both the left- and right-hand portions indicate colonization.

**Figure 2 plants-12-01248-f002:**
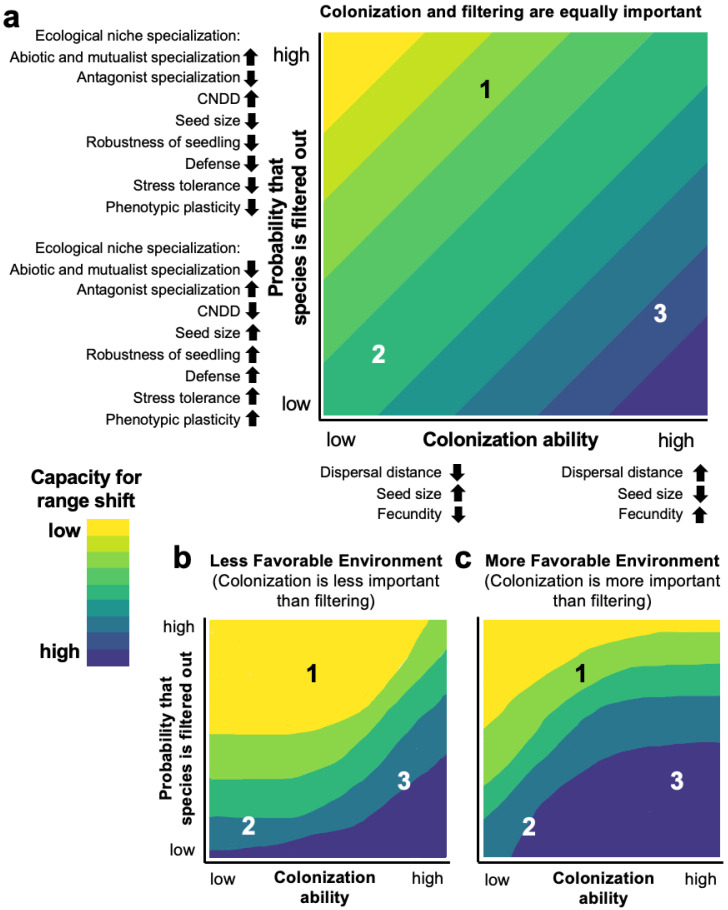
A plant species’ capacity for range shifts is shaped by the relationship between its colonization ability and its probability of being filtered out of a site. The capacity for plant range shifts is determined equally by the effects of colonization ability and the probability that a species experiences habitat filtering (i.e., is filtered out of a site) after colonization in (**a**), is more limited by a species’ probability of being filtered out in (**b**), and is more limited by a species’ colonization ability in (**c**), with low (yellow) to high (purple) capacity for range shift. Trade-offs in species’ ecological strategies influence their colonization ability and their probability of being filtered out of a site through a phenotype–environment mismatch. Trait values that should increase or decrease colonization ability or probability of being filtered out are denoted on the high and low ends of each axis and apply to all three graphs, even though they are only detailed in (**a**). The numbers 1, 2, and 3 represent the positions of three hypothetical species positioned on the x and y axis coordinates based on their trait values. Species 1 would have a moderate dispersal distance and fecundity and small to moderately sized seeds; Species 1 could represent an abiotic condition and mutualist specialist and antagonist generalist, experience high pressure from conspecific negative density-dependence (CNDD), produce less robust seedlings, be a less well-defended and stress tolerant individual, and/or exhibit low phenotypic plasticity. Species 2 would have low seed dispersal distance and low fecundity and would be a large-seeded species; Species 2 could represent an abiotic condition and mutualist generalist and antagonist specialist, experience minimal effects of CNDD, produce more robust seedlings, be a well-defended and stress tolerant individual, and/or exhibit high phenotypic plasticity. Species 3 would have high colonization ability, due to having a longer dispersal distance, smaller seed size, and higher fecundity, and similar traits on the *y*-axis as Species 2. In (**a**), colonization ability and the probability of being filtering out are assumed to be equally important in determining the capacity for range shift (i.e., additive effects), leading to linear contours in plant range shift capacity. In this scenario, Species 1 has a low–moderate capacity for range shift because it has a moderate colonization ability but high probability of being filtered out; Species 2 has a moderate capacity for range shift, due to its low colonization ability and low probability of being filtered out; and Species 3 has a high capacity for range shift, due to a high colonization ability and relatively low probability of being filtered out. Both (**b**,**c**) represent environments where there are interactive effects between colonization ability and the probability of being filtered out, leading to curved contours. In (**b**), the environment is weighted by the probability of being filtered out, making it a stronger limiting factor than colonization ability on capacity for range shift, resulting in concave-up contours in plant range shift capacity. In this scenario, Species 1 will be unlikely to expand its range, while Species 2 and 3 should both be able to expand their ranges. In (**c**), the environment is weighted by colonization ability, making it a stronger limiting factor than the probability of being filtered out on the capacity for range shift, resulting in concave-down contours in plant range shift capacity. In this scenario, compared to (**b**), Species 1 has a better capacity for range shift, because the environment is more favorable, and Species 2 and 3 both have high capacities for range shift despite strong differences in colonization ability. There will not be any species located in the top left-most corner of the graphs with very low colonization ability and very high probability of being filtered out, as species with this strategy would have a very low capacity for range shifts and would not survive in their current distributions. Similarly, there should be relatively fewer species in the top right-most corner of the graphs that have high colonization ability and high probability of being filtered out, as species with these strategies would face challenges in establishing a viable population (due to high filtering) and would have below-moderate capacity for range shifts. See Table 1 and Section 6 text for further explication of trade-offs in species’ ecological strategies, and how these operate to increase or decrease the capacity of a species to shift its habitat and/or geographic range.

## Data Availability

No new data were created or analyzed in this study. Data sharing is not applicable to this article.

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
