# Peer review of "Plant Species’ Capacity for Range Shifts at the Habitat and Geographic Scales: A Trade-Off-Based Framework"

_plants, 2023, doi:10.3390/plants12061248_

Round 1
Reviewer 1 Report
Dear Editor, The manuscript named"Plant species' capacity for range shifts at the habitat and geographic scales: A trade-off-based framework" provides a very comprehensive review of the ability of plants to shift their geographic range and habitat in a period like the current one characterized by rapid climate change. The paper is well written from a linguistic point of view and quite complete in dealing with the various topics. Therefore I believe it can be published in its current form, with only a few small spelling corrections that I have highlighted in the attached file.
Regards

Author Response
Comments from Reviewer #1
- Dear Editor, The manuscript named "Plant species' capacity for range shifts at the habitat and geographic scales: A trade-off-based framework" provides a very comprehensive review of the ability of plants to shift their geographic range and habitat in a period like the current one characterized by rapid climate change. The paper is well written from a linguistic point of view and quite complete in dealing with the various topics. Therefore I believe it can be published in its current form, with only a few small spelling corrections that I have highlighted in the attached file.
We thank Reviewer #1 for the encouraging assessment of our manuscript and have made the grammatical corrections noted in the PDF on the revised version of our manuscript.
Reviewer 2 Report
Dear Authors!
General Comment
Manuscript is well written, with extensive use of literature. My general comments goes to the fact that all written goes well for plant species without human mediated dispersion across continents and otherwise natural geographical barriers. Latter opening a numerous additional issues with respect to the alien and invasive plant species. Including those, would make such review paper even more complex, with number of needed references to add in huge numbers. Therefore, I do not think that this should be done. Instead, I think it should be mentioned in the manuscript (maybe even in the title) clearly stating that the proposed framework is for “naturally” dispersed species.
Specific Comments
Figure 2. – Caption – I am not sure whether it is a result of miss-formatting or intentionally, but I found length of the Caption unsuitable. It is actually a part of the manuscript, given the extent of it and style how it is written. Therefore, please reformat this part. Furthermore, in Line 281, it should be “Species 2 and 3” instead of “Species B and C”.
Lines 453-454 – “latitude” and “climate” is kind of redundant in terms of their impact on shaping the environmental conditions
Line 509 – “aseasonal habitat” is term that is not clear to me. Can you please explain it, a bit more.
Lines 545-548 – Please consider rephrasing this sentence. Although citation format of the literature is with numbers, in this case you can add Authors to make it clear that these “19 considered tree species” were considered by these Authors, and not overall by Yourself in Your manuscript. Something like: “In temperate forest, for both saplings and adults, for most of the 19 considered tree species Purves (37) had found differences in growth…” or similar.
Line 604 – do you mean “grassland species” here, since Tragopogon dubius does not belong to the Poaceae (grass family)?
Line 841- missing “be” in “that can be used in …”?
Author Response
Comments from Reviewer #2
Dear Authors!
General Comment
- Manuscript is well written, with extensive use of literature. My general comments goes to the fact that all written goes well for plant species without human mediated dispersion across continents and otherwise natural geographical barriers. Latter opening a numerous additional issues with respect to the alien and invasive plant species. Including those, would make such review paper even more complex, with number of needed references to add in huge numbers. Therefore, I do not think that this should be done. Instead, I think it should be mentioned in the manuscript (maybe even in the title) clearly stating that the proposed framework is for “naturally” dispersed species.
We thank Reviewer #2 for bringing up this important topic and agree that this is an excellent point. The reviewer is correct that the main focus of our paper was not on human-mediated dispersal of plants that have the potential to become invasive species into non-native geographic areas. To clarify that the proposed framework is for naturally dispersed species, and to briefly outline the parallels in phenotype-environment matching processes between naturally dispersed plants and human-introduced species, we have added a few points and key citations pertaining to invasive species throughout the text in the Section 1: Introduction (Lines 137-140) and in Section 5: Phenotype-environment matching (Lines 453-458, 656-664, 719-721, 753-754).
Specific Comments
- Figure 2. – Caption – I am not sure whether it is a result of miss-formatting or intentionally, but I found length of the Caption unsuitable. It is actually a part of the manuscript, given the extent of it and style how it is written. Therefore, please reformat this part. Furthermore, in Line 281, it should be “Species 2 and 3” instead of “Species B and C”.
We thank Reviewer #2 for catching our error in writing Species B and C in the Figure 2 caption and have changed this to say Species 2 and 3. However, we have otherwise decided to retain the length and content of the Figure 2 legend. This legend is very long because Figure 2 is the key conceptual figure underpinning our review paper, and therefore the legend is critical in providing the reader with the complete explanation needed to fully understand the figure independent of the text. We deliberated in moving some of the figure legend text into Section 6: Plant species’ capacity for range shifts but felt that dividing the legend into sections would be confusing and create more problems with flow than advantages. However, if the editors feel that the legend length is an issue, we will move it.
- Lines 488-490 – “latitude” and “climate” is kind of redundant in terms of their impact on shaping the environmental conditions
We agree with this comment and have deleted “latitude” in the manuscript.
- Line 550 – “aseasonal habitat” is term that is not clear to me. Can you please explain it, a bit more.
We regret the poor wording of this sentence, which was meant to be an illustrative example of specialist plant species’ enhanced performance in their home habitat type compared to generalist species. We have reworded this sentence as follows: “For example, a reciprocal transplant experiment evaluating seedling performance along a climatic gradient found higher survival and growth of specialists than widespread generalist species when planted in their home habitat”.
- Lines 588-590 – Please consider rephrasing this sentence. Although citation format of the literature is with numbers, in this case you can add Authors to make it clear that these “19 considered tree species” were considered by these Authors, and not overall by Yourself in Your manuscript. Something like: “In temperate forest, for both saplings and adults, for most of the 19 considered tree species Purves (37) had found differences in growth…” or similar.
We agree that this sentence was misleading in its wording, and to clarify we are reporting results from a previously published empirical study and that this findings are not being reported in our review, we have rephrased the sentence as follows: “A study of 19 tree species in a North American temperate forest found differences in growth rate for populations at their range limit versus range center, with growth rates generally being lower at the colder northern limit and higher at the warmer southern limit”.
- Line 670 – do you mean “grassland species” here, since Tragopogon dubiusdoes not belong to the Poaceae (grass family)?
We are grateful that the reviewer caught this mistake and have revised this line to read “grassland species” instead of “grass species”.
- Line 923- missing “be” in “that can be used in …”?
We are grateful that the reviewer caught this grammatical error and have revised this line to read “that can be used in…”.
Reviewer 3 Report
This article deals with the influence of climate change on the distribution of plant species, with species moving towards more suitable ecological niches. The authors state the dispersion of plant phenotypes towards optimal ecological niches, but point out the possibility that the adaptation time of the species is insufficient given the high rate of climate change, with phenotypic plasticity being crucial in the adaptation of the species.
It is a magnificent work on the influence of climate change on the distribution of species and their adaptation. But it is poorly structured, the authors make a single introduction section; it is not possible to differentiate between introduction, methodology, results, discussion and conclusions. The article must be completely restructured.
Author Response
Comments from Reviewer #3
- This article deals with the influence of climate change on the distribution of plant species, with species moving towards more suitable ecological niches. The authors state the dispersion of plant phenotypes towards optimal ecological niches, but point out the possibility that the adaptation time of the species is insufficient given the high rate of climate change, with phenotypic plasticity being crucial in the adaptation of the species.
It is a magnificent work on the influence of climate change on the distribution of species and their adaptation. But it is poorly structured, the authors make a single introduction section; it is not possible to differentiate between introduction, methodology, results, discussion and conclusions. The article must be completely restructured.
We thank Reviewer #3 for their positive assessment of our manuscript. However, as our manuscript is a review paper rather than a data-driven paper, the traditional sections used in a data-driven paper (e.g., introduction, methods, results, discussion) do not apply in this case because the paper did not involve any data collection. Therefore, we have elected to keep the content and order of the sections as is in the revised manuscript.